# Selenium Compounds as Novel Potential Anticancer Agents

**DOI:** 10.3390/ijms22031009

**Published:** 2021-01-20

**Authors:** Dominika Radomska, Robert Czarnomysy, Dominik Radomski, Krzysztof Bielawski

**Affiliations:** Department of Synthesis and Technology of Drugs, Medical University of Bialystok, Jana Kilinskiego 1, 15-089 Bialystok, Poland; dominika.radomska@umb.edu.pl (D.R.); d.radomski@post.pl (D.R.); kbiel@umb.edu.pl (K.B.)

**Keywords:** selenium compounds, cancer, anticancer agents, molecular target, drug resistance, combinatorial therapy, chemoprevention

## Abstract

The high number of new cancer incidences and the associated mortality continue to be alarming, leading to the search for new therapies that would be more effective and less burdensome for patients. As there is evidence that Se compounds can have chemopreventive activity, studies have begun to establish whether these compounds can also affect already existing cancers. This review aims to discuss the different classes of Se-containing compounds, both organic and inorganic, natural and synthetic, and the mechanisms and molecular targets of their anticancer activity. The chemical classes discussed in this paper include inorganic (selenite, selenate) and organic compounds, such as diselenides, selenides, selenoesters, methylseleninic acid, 1,2-benzisoselenazole-3[2H]-one and selenophene-based derivatives, as well as selenoamino acids and Selol.

## 1. Introduction

The chemistry of selenium (Se) compounds is not a young field, since its beginning was in the first half of the 19th century. The first steps in this scope were initiated in 1836 by the then toxicologists, who discovered a Se metabolite, diethyl selenide, excreted by inhalation during research using inorganic selenium compounds on animals. Meanwhile, the first synthesis of selenium-containing (Se-containing) compounds took place in 1847, when ethylselenol was obtained [1]. Since then, selenium chemistry has developed greatly and investigators have started to look at whether it can help with society’s health problems, including cancer. The high number of new cancer incidences and the associated mortality continue to be alarming [2], leading to the search for new therapies that would be more effective and less burdensome for patients. As there is evidence that Se compounds can have chemopreventive activity [3,4,5], studies have begun to establish whether these compounds can also affect already existing cancers. This review aims to discuss the different classes of Se-containing compounds, both organic and inorganic, natural and synthetic, and the mechanisms and molecular targets of their anticancer activity. The chemical classes discussed in this paper include inorganic (selenite, selenate) and organic compounds such as diselenides, selenides, selenoesters, methylseleninic acid, 1,2-benzisoselenazole-3[2H]-one and selenophene-based derivatives, as well as selenoamino acids and Selol.

These compounds are metabolized into different redox-active products, and thus their metabolites belong to two pools: a hydrogen selenide (e.g., selenite) and a methylselenol (e.g., methylseleninic acid, selenoesters) pool, which exhibit a slightly different mechanism of action and toxicity, and their properties depend on the doses used [6,7,8]. Furthermore, the high reactivity of these metabolites results from their increased nucleophilic character, which ensures their anticancer efficacy [8]. In the course of the studies, it was found that methylselenol metabolites exhibit multiple beneficial chemopreventive as well as anticancer effects, whereas an excess of hydrogen selenide metabolites in a cell may cause single-strand DNA breaks (SSBs) [7]. For this reason, it is thought that organic Se compounds are less toxic than inorganic forms that contain Se [6,9]. The anticancer mechanism of Se-containing compounds is mainly based on inducing apoptosis in cells [10], but apart from that, these compounds also affect gene expression and various cell signal pathways, DNA repair/damage, as well as the angiogenesis process or metastasis [11].

In addition, it was observed that these compounds sensitize cancer cells to standard chemotherapy/radiotherapy and synergistically increase their effectiveness and may also reduce their side effects [6,12]. Unfortunately, these mechanisms are multidirectional and are not fully understood [7]. The mechanisms described above mainly arise from the high redox activity of metabolites, which is associated with the formation of reactive oxygen species (ROS) and the oxidation of protein thiol groups in the cell, leading to oxidative stress [11,13]. However, Se and its compounds are like a “double-edged sword”; depending on the dose, they may have antioxidant or prooxidant properties. At nutritional levels, redox-active Se-containing compounds have an antioxidant activity only after the incorporation of Se into selenoproteins, exhibiting a chemopreventive effect, whereas in supranutritional doses, they manifest their prooxidant and anticancer properties [13]. Finally, it is also worth mentioning that Se compounds may trigger other types of cell death apart from apoptosis, which may proceed via both internal and external pathways. Non-apoptotic events may also occur, such as cell cycle arrest [3,7,12], necrosis [3,7], autophagy [7,12], ferroptosis [14], necroptosis [13], entosis [15], anoikis [8,16], NETosis [17], or mitotic catastrophe [7]. Among the types of cell death caused by Se compounds, ferroptosis, which was first described by Dixon and Stockwell in 2012, seems to be a very interesting process [18,19]. It is non-apoptotic regulated iron-dependent cell death with the simultaneous growth of ROS, which leads to the accumulation of lipid peroxides in the membrane [14,18,19,20,21]. Its basic mechanism results from the imbalance between the production of oxidizing substances (ROS) and cellular antioxidant systems [18,19,21], which causes an increased level of ROS inside the cell and oxidative stress [18]. It turns out that the catalyst of this type of cell death is iron, which accumulates in the cell and then undergoes the Fenton reaction and generates free radicals’ formation. Moreover, this element also increases the activity of the enzyme—lipoxygenase, which also takes part in lipid peroxidation [19,20]. The main enzyme that prevents the development of ferroptosis and is its key regulator is Se-dependent glutathione peroxidase 4 (Gpx4, phospholipid hydroperoxide glutathione peroxidase, PHGPx) [19,21,22]. Therefore, its expression and activity are determined by the levels of this trace element in the body (too low Se concentration limits them). This is due to the fact that in the active site of Gpx4, there is a selenocysteine residue, which, together with glutathione (GSH), is responsible for the reduction of lipid peroxides to the corresponding alcohols, which prevents the formation and accumulation of ROS [19,20,21,22,23]. Gpx4 alone cannot exist without the substance that regenerates it, which is why GSH, which is responsible for the reduction of oxidized Gpx4, is also necessary to prevent ferroptosis [20,21]—both of these components are very important for the antioxidant protection of the cell [14,19]. In summary, ferroptosis can occur when the cell’s antioxidant defense is exceeded (excessive formation of ROS/lipid peroxides by too high iron levels and other reactions leading to their formation) or when the antioxidant system is weakened (decrease in expression/inactivation/depletion of Gpx4 or depletion/decrease in GSH levels) [14,18,19,20,21]. Possible types of cell death induced by Se-containing compounds are presented in Figure 1.

## 2. Inorganic Selenium Compounds

Both **selenite** (SeO_3_^2−^, SeL, Figure 2, structure 1) and its sodium salt, **sodium selenite** (disodium selenite, Na_2_SeO_3_, SS, Figure 2, structure 2), belong to a group of inorganic compounds, which were tested as the first Se-containing compounds, and the scope of research on their anticancer properties was very extensive [3,6,7].

During the metabolism of SS in vivo, the formation of hydrogen selenide (H_2_Se) occurs [5,24], which is then methylated forming methylselenol [25,26]. The in vivo study concluded that SeL (3 mg/kg body weight (b.w.)) was well tolerated but was genotoxic [27], while at a concentration of about 127 µM Se, it was toxic and genotoxic to primary human keratinocytes (NHK) [28]. This is due to the fact that medium (3–5 ppm; 0.3–0.5 mg Se/kg b.w.) or dietary (0.1 ppm; 0.01 mg Se/kg b.w.) doses of SeL are metabolized to selenide (Se^2−^), which is then incorporated into selenoproteins [3,29,30]. In turn, higher concentrations (>5 ppm; >0.5 mg Se/kg b.w.) of this compound lead to its reaction with oxygen and the formation of ROS, which causes oxidative stress, and this ultimately results in the cytotoxic and genotoxic effects [3,28,30,31]. Selenate (Figure 2, structure 3) also exhibits genotoxic properties [27]. For this reason, chronic high doses of SeL should not be ingested, as its genotoxic properties in normal cells may lead to their mutagenesis and paradoxical cancer development [27]. It is worth quoting here the results of a study in which a combination of SeL (126.6 µM Se) and Trolox (80 µM; water-soluble Vit. E) increased the prooxidative and genotoxic effects of SeL on NHK cells, which could have resulted from the formation of vitamin E radicals by SeL. No such results were observed when using selenomethionine (SeMet) + Trolox, which proves that SeL is a prooxidant [28]. The anticancer effects of SeL and SS were observed on many cancer cell lines, including prostate [24,25,26], breast [32], lung [33], hepatoma [34], bladder [35], and glioblastoma [9] or osteosarcoma [36].

Furthermore, cells resistant to the available cytostatic agents in comparison with cells sensitive to them are more susceptible to SeL [6]. It was reported that SeL affects and disrupts intracellular GSH levels. This is because SeL is reduced to selenide (Se^2−^) by GSH, resulting in the rapid depletion of GSH in the cell [34]. The selenide produced as a result of this reaction reacts with oxygen, leading to ROS formation [30]. Since GSH is too low in the cell, there is a redox imbalance and excessive amounts of ROS form, which in turn induces oxidative stress [30,33,34,35] and other signal pathways, and leads to apoptosis [7]. In a study by Shigemi et al. [37], excessive amounts of ROS caused ER stress and subsequent apoptosis via activating the unfolded protein response (UPR) pathway as a result of SS activity.

However, in the apoptosis induced by SeL, mitochondria are the main contributor [7,34]. SS easily diffuses into the mitochondria [30], reacting with the mitochondrial GSH and producing superoxide, which reacts with the components of this cellular organelle [38]. Then, the oxidation of protein thiol groups occurs. Both the generated superoxides and the oxidized thiols open the pores of the mitochondrial permeability transition (MPT) [34,38]. At this point, the potential of the mitochondrial membrane (ΔΨm) decreases [30,34,35,38] and cytochrome c is released into the cytoplasm [3,38]. In the case of selenate, this apoptotic pathway does not occur [30,38].

In addition, the p53 protein is involved in the apoptotic pathway. As SeL leads to DNA fragmentation (DNA fragments 50–300 bp) and DNA SSBs [3], a signal of DNA damage appears, which leads to p53 activation [24,31]. The activation of p53 increases the levels of the proapoptotic protein, Bax [24,36], which is then translocated into the mitochondria, oligomerizes, and finally destabilizes the mitochondrial membrane [24]. Furthermore, the introduction of cells into the apoptotic pathway may be caspase-mediated [25,26,30], but it may also be a caspase-independent process [7,24].

To sum up the above-described mechanisms, as a result of SeL metabolism, excessive generation of ROS is observed, which results in DNA SSBs and activation (phosphorylation) of p53, which in turn induces increased transcription of Bax and then its (Bax) translocation to the mitochondria. The membrane is destabilized (opening of MPT pores) and cytochrome c is released to cytosol, which activates caspases and finally leads to cancer cell apoptosis. The caspase-independent apoptosis occurs in cells without a functional p53 protein (e.g., prostate cancer PC-3 or DU145 cells), but is not as efficient as caspase-dependent apoptosis (no p53-mitochondria feedback loop, which sustains ROS production) [24]. As can be seen, apoptosis occurs through the internal (mitochondrial) pathway, but there are reports of a component of the external pathway through the DR5 death receptor. This was observed in DU145 prostate cancer cells, where SeL induced the upregulation of the DR5 receptor and simultaneous activation of the internal pathway, which may lead to the assumption that the internal and external pathways may be related [3].

Moreover, the inhibition of cancer cell proliferation by SeL may occur by suppressing NF-κB DNA binding [25], inhibiting protein kinase C (PKC) [7,29] or histone deacetylases (HDAC) [6,32] activity, as well as poisoning topoisomerase II (by reacting with thiols incorporated into this enzyme) [31] or by affecting other molecular targets, i.e., p38 mitogen-activated protein kinase (p38MAPK), c-Jun NH2-terminal kinase (JNK), AP-1 [3,7,26], ATM [31], AKT [7,26]. As far as the cell cycle is concerned, SeL treatment led to cell cycle arrest in phase S and irreversible inhibition of cell proliferation, while selenate arrested the cycle in phase G2, and growth inhibition was reversible [26,30]. So far, the impact of SeL on apoptotic events in cancer cells has been described. Apart from the apoptotic phenotype, there are several other non-apoptotic types of cell death, i.e., necrotic or autophagic phenotype, which had also been observed during SeL treatment [7]. In the study carried out by de Miranda et al. [32], the authors observed that SeL induced necrosis in breast cancer MCF-7 cells, while Soukupová et al. [35] found autophagy and necrosis in chemoresistant human bladder cancer RT-112/D21 cells. In addition, Subburayan et al. [14] reported in their paper that SS induced ferroptosis in human cancer cells such as breast (MCF-7), prostate (PC-3), and glioma (U-87MG).

In addition to the research on the effect of SeL on cancer cells, there were also studies that used combination therapy of SeL with standard chemotherapy. Thus, it was proved that SeL synergistically enhanced the effectiveness of imatinib against colorectal cancer HTC116, cisplatin against MCF-7 cells [39], or TrxR (thioredoxin reductase) inhibitors (auranofine, ethaselen) against human lung and ovarian cancer cells [6]. Furthermore, it was reported that apart from synergistic enhancement, SeL also reduced the side effects (e.g., toxicity) of commonly used cytostatic drugs, among other (a.o.) 5-fluorouracil (5-FU) or docetaxel [6]. There are also human studies, in which the combination of SS with commonly used cancer therapies was applied. Table 1 shows clinical trials of such combined therapies and the study using SS alone.

In conclusion, during the use of high doses of SeL, the selenide (Se^2−^) formed during metabolism is responsible for the genotoxic effects resulting from the formation of large amounts of ROS, which interact with DNA. On the other hand, methylation of Se^2−^ leads to obtaining a compound (methylselenol) that does not show this property. This has led to the design of stable compounds, which are precursors (“prodrugs”), releasing anticancer methylselenol during their metabolism [25]. These methylselenol precursors will be discussed in the following section of this review.

## 3. Organic Selenium Compounds

### 3.1. Diselenides

Diselenides are a class of compounds that contain the Se-Se bond in their structure and have the general chemical formula R_2_Se_2_. **Dimethyl diselenide** (Figure 3, structure 5) is a compound found in nature that exhibits antioxidant properties and strongly induces NADPH quinone oxidoreductase, whereas **dipropyl diselenide** (Figure 3, structure 6) and **dibutyl diselenide** (Figure 3, structure 7) are known for their prooxidant effects, even at low concentrations [46,47].

In the case of **diphenyl diselenide** (Figure 3, structure 8), it has been characterized as an antioxidant and inhibitor of nociception, which also protects neurons from damage and has antidepressant properties [48]. Its protective activity on cells was tested on murine J744 macrophage-like cells and during tamoxifen (TAM) therapy on mammary gland cancer MCF-7 cells. In the first case, after the generation of ROS and subsequent treatment with compound **8**, there was a decrease in NF-κB expression in the cells, while in the second study, it counteracted oxidative damage in cells caused by TAM without affecting its activity. At low doses, it had protective properties against genotoxic substances.

Meanwhile, in vivo studies revealed that diphenyl diselenide can be toxic, depending on its route of administration and dosage [46]. Mainly, this compound was studied for its antioxidant activity, while there is less research on its cytotoxic effects on cancer cells. The inhibitory activity of compound **8** on neuroblastoma (SH-SY5Y) cell growth seems to be mediated by the extracellular signal-regulated kinase (ERK 1/2) pathway, which was presented in a paper by Posser et al. [49]. Additionally, in the study by Nedel et al. [48], compound **8** at a concentration of 80 µM exhibited cytotoxicity and apoptosis induction in HT-29 cells, which is consistent with the findings of Posser et al. [49] that diphenyl diselenide leads to cell death via the apoptosis pathway.

In turn, **4,4′-dimethoxy-** and **3,3′-ditrifluoromethyl-diphenyl diselenide** (Figure 3, structures 9 and 10) were more cytotoxic than **8** to HT-29 cells and induced apoptosis at 20 µM, out of which **9** caused a higher percentage of this cellular event. It was found that compounds **9** and **10** induced apoptosis via the caspase-dependent (↑ AIF, ↑ caspase 9 and 8) and caspase-independent (↑ Bax, ↓ survivin) pathways, and arrested the cell cycle in the G2/M phase via genes p53, p21 (↑ expression), and MYC (↓ expression) [48].

Studies on the cytotoxicity of **3′,5′,3,5-tetratrifluoromethyl-diphenyl diselenide** (Figure 3, structure 11) were carried out on several cancer cell lines, i.e., HL-60 (human leukemia, IC_50_ = 8 µM), PC-3 (prostate cancer, IC_50_ = 13 µM), MCF-7 (breast cancer, IC_50_ = 18 µM), MIA-PA-Ca-2 (pancreatic cancer, IC_50_ = 25 µM) and HCT-116 (colorectal cancer, IC_50_ = 27 µM). Compound **11** induced apoptosis via the internal pathway (mitochondrial) and arrested cells in phase S of the cell cycle. A simulation of molecular docking of diselenide **11** demonstrated that this compound selectively binds to the minor groove of DNA via hydrogen bonds in which fluorine atoms are involved. Additionally, there are hydrophobic interactions between the benzene rings and DNA.

**1,2-bis(chloropyridazinyl) diselenide** (Figure 3, structure 12) had the highest growth inhibition of MCF-7 cancer cells among all 17 compounds synthesized and studied by Kim et al. [50]. Proliferation inhibition by this compound was dose-dependent and its IC_50_ amounted to 10.34 µM.

**N,N’-((diselanediylbis(2,1-phenylene))bis(methylene))di(ethaneamine)** (Figure 3, structure 13), which was synthesized by Krasowska et al. [51], exhibited the inhibition of cell proliferation in micromolar concentrations in three cell lines of cervical (HeLa, IC_50_ = 20 µM) cancer, breast (MCF-7, IC_50_ = 30 µM) cancer, and leukemia (K652, IC_50_ = 15 µM). It was found that **13** is an inhibitor of glutathione-S-transferase (GST)—this enzyme is responsible for increased drug removal in cancer cells. Moreover, the use of this compound with cisplatin enhanced its cytotoxic activity, possibly resulting from GST inhibition. The authors also conducted a molecular docking of **13** in a crystal structure of GST in their study. It turned out that it can be located in the H-site of the active center of the enzyme and be stabilized by many interactions. It was also found that the key interactions for the inhibition of GST activity by compound **13** are π–π stacking interactions with Y108 and the diselenide bridge docked near Y7 (a phenol group).

The **indol-containing diselenide** (Figure 3, structure 14) and **bis(4-amino-3-carboxyphenyl) diselenide** (Figure 3, structure 15) affect the kinases. The activity of cyclin kinase (CDKs) is slightly suppressed by **15**, while **14** inhibits the tyrosine kinase in cells [46].

In vitro studies of diselenides suggest that they may be promising candidates for anticancer agents, but their application, including, among others, that of diphenyl diselenide, may be limited due to their toxicity in vivo. The toxic effects of these compounds depend on the dosage, vehicle, and route of administration, as well as the age or species of the tested animal [52]. It was observed that diphenyl diselenide was less toxic in rats than in mice—the LD_50_ of this compound was 374.4 mg/kg in rats and 65.52 mg/kg in mice after intraperitoneal administration [52,53]. Acute subcutaneous (<156 mg/kg) and oral exposure to diphenyl diselenide and its chronic administration (<31.2 mg/kg, subcutaneously) did not cause toxic effects in mice. In contrast, repeated administration at >31.2 mg/kg subcutaneously and acute intraperitoneal exposure to this compound had toxic effects in these rodents [52,53,54]. Oral administration of 30 mg/kg (supranutritional dose of diphenyl diselenide) for 8 months was non-toxic in rabbits, whereas acute intraperitoneal administration of 1.56 and 15.6 mg/kg was hepatotoxic to them, while a dose of 156 mg/kg was lethal to 85% of the population [52]. The main mechanism responsible for the toxic properties of diselenides is their prooxidant activity, which involves interaction with functionally relevant -SH (thiol) groups of proteins, leading to their oxidation and loss/decrease in their function [53,55,56]. Depletion of intracellular GSH is one of these effects [53,55]. Furthermore, by the same mechanism, these compounds can inhibit the enzyme δ-aminolevulinic acid dehydratase (δ-ALA-D) in the blood, kidney, liver, and brain (highly lipophilic molecules that can cross the blood–brain barrier (BBB)) [52,53,55,56]. Inhibition of this enzyme in the blood results in the accumulation of aminolevulinic acid (ALA), characterized by some prooxidant properties [53]. In turn, the crossing of the BBB by diselenides results in their neurotoxic effects [52,56]. This is manifested by the induction of convulsions in mice after the administration of diphenyl diselenide. It is noteworthy that this effect was observed only in intraperitoneal administration, indicating that this toxicity depends on the route of administration. This may be due to the accumulation of the intermediate metabolite of diphenyl diselenide in the brain, although it seems more possible that inhibition of δ-ALA-D leads to accumulation of ALA, which has a proconvulsant effect. Interestingly, the neurotoxicity of diphenyl diselenide is also determined by the age of the tested animal. Oral administration of this compound induced seizures in rat pups, but not in adults. In addition, glutamatergic system impairment was found [52,53,56,57]. In the case of 4,4′-dimethoxy- and 3,3′-ditrifluoromethyl-diphenyl diselenide, reduction/abolition of the occurrence of convulsions in mice was observed [52,57]. The acute oral toxicity of 3,3′-ditrifluoromethyl-diphenyl diselenide was slightly higher (LD_50_ > 278 mg/kg) and 4,4′-dimethoxy-diphenyl diselenide was slightly lower (LD_50_ > 372 mg/kg) than that of diphenyl diselenide (LD_50_ > 312 mg/kg). This means that the introduction of substituents into the aromatic ring of diphenyl diselenide had no significant effect on reducing its toxicity. Nevertheless, 4,4′-dimethoxy-diphenyl diselenide did not inhibit δ-ALA-D and did not cause death in experimental animals—the compound with the -CF_3_ group exhibited both of these properties [57]. However, another member of the diselenide class, 2,2′-dithienyl diselenide, at a single oral dose of 100 mg/kg exhibited systemic toxicity in rats and caused death. In this study, inhibition of δ-ALA-D, an increase in aspartate (AST) and alanine aminotransferase (ALT) activities, and a decrease in urea levels were observed, with no effect on creatinine levels, showing that the compound had a hepatotoxic effect but no effect on renal function [56]. A very important finding is that in studies using diselenides-loaded nanocapsules, no toxic effects on mice were shown, which gives hope for a reduction in their harmful properties and possible application in human trials in the future [55,58]. Summarizing the toxicological data of diselenides in in vivo studies, more information from further animal studies on them is needed to unambiguously determine the exact mechanism of their harmful effects on the body, as well as methods to prevent them. Although the synthesis of diphenyl diselenide derivatives proved to be not entirely successful (their toxicity was comparable to that of diphenyl diselenide), the reduction in toxic effects may be probably achieved by the formulation of the administered compound.

### 3.2. Selenides

Another class of selenoorganic compounds is selenides (also called selenoethers), with the general formula R-Se-R. Among them, there are derivatives with chemopreventive properties, involving the binding of heavy metals or Gpx-like (glutathione peroxidase-like) activity. Meanwhile, the anticancer activity of the compounds was tested on various cancer cell lines, including colon, uterine, lung, liver, and breast cancer. **Zidovudine derivatives** (active at medium micromolar concentrations) induced apoptosis through the internal (mitochondrial) pathway, while **phenylindolyl ketone derivative** was active at nanomolar concentrations, causing the arrest of the cell cycle in the G2/M phase, which led the cells to the apoptosis pathway as a result of a decrease in mitochondrial membrane potential (ΔΨm) and additionally inhibited tubulin polymerization [46].

Another selenide compound, which is also an inhibitor of tubulin polymerization, is a **combretastatin 4-A analog** (Figure 4, structure 16). This analog was tested on a panel of four cancer cell lines, exhibiting an inhibitory effect on cell growth in nanomolar concentrations, and its activity was stronger than **combretastatin A-4** (CA-4, Figure 4). IC_50_ values for this compound for breast (MCF-7), kidney (768), colon (HT-29), and prostate (PC-3) cancer were 10, 680, 280, and 80 nM, respectively. In the molecular docking simulation of compound **16**, it was found that it binds at the colchicine-binding site of tubulin. The selenoether angle was 101.19°, which provides deep receptor binding in tubulin (colchicine-like). Between the oxygen atom of the methoxy substituent in the para position (ring A, Figure 4) and the thiol from the side chain of the residue b-Cys 241 of the receptor site of the enzyme, the hydrogen bond forms. Another hydrogen bond binds the oxygen atom from the para-methoxy substituent **of ring B (Figure 4)** with the amide nitrogen of residue a-Val181 at the receptor site of the enzyme. As a result of this mechanism, tubulin is inhibited and its enzymatic activity decreases [59].

### 3.3. 1,2-Benzisoselenazole-3[2H]-One Derivatives

**Ebselen** (2-phenyl-1,2-benzisoselenazol-3(2H)-one, 2-phenyl-1,2-benzoselenazol-3-one, also called PZ 51, DR3305, SPI-1005, Figure 5, structure 17) is a Se-containing heterocyclic compound that shows chemopreventive properties through its anti-inflammatory and antioxidant activity. This is possible due to its weak GPx-like action [3,60]. As a result of ROS scavenging, this compound prevents the development of cellular oxidative stress, protecting against the oxidation of cell components and the formation of DNA mutations [60]. Yang et al. [61] demonstrated in their study that treatment with 25 µM ebselen reduced the effect of H_2_O_2_ on growth inhibition, DNA damage, and lipid peroxidation in HepG2 cells. Among the wide range of studies on the antioxidant properties of ebselen, there are also reports of its possible inhibitory effects on proliferation. In experiments on HepG2 cells, this compound at a dose of 50–75 µM intensified the apoptosis process. This was caused by the oxidation of thiols present inside the cell, which resulted in their sharp depletion [3].

In another study, ebselen inhibited the proliferation of human multiple myeloma cells, increasing their apoptosis, which was associated with high levels of ROS and their impact on mitochondria. After treating cells with 40 µM ebselen for 4 h, translocation of Bax to the mitochondria, a decrease in their membrane potential (ΔΨm), and a final release of cytochrome c into the cytoplasm were observed [60]. In pancreatic cancer xenografts in mice, ebselen at a dose of 160 and 640 µg/day (no differences in tumor size between doses) reduced tumor development by 58% [8]. In addition, other investigators showed that co-treatment of ebselen with tumor necrosis factor α (TNF-α) in glioblastoma cells resulted in increased sensitivity to TNF-α and increased apoptosis. The observed effect resulted from the activation of two different pathways, which included the deactivation of NF-κB and stimulation of death-inducing signaling complex (DISC) production, which finally led to an activation of caspase 8, triggering the executive caspase cascade and induction of apoptosis [12,62]. An in vitro study using the MCF-7 cell line showed that this compound at a concentration of 25 µg/mL combined with γ-radiation at 6 Gy resulted in the inhibition of proliferation and induction of apoptotic cell death, which resulted from the regulation of pro- and anti-inflammatory cellular response and expression of genes and proteins involved with its pathways [63].

In the case of human studies, there are two clinical trials on ebselen. Both of them are randomized, quadruple-blind (participant, care provider, investigator, outcomes assessor), placebo-controlled, and parallel assignment. The first trial (NCT number: NCT01452607) is in phase 1 and is investigating the pharmacokinetic profile of ebselen within 24 h (dose: 200 mg) and its safety (time frame: 1 month, dose range: 200–1600 mg) in healthy subjects [64]. The second clinical trial (NCT number: NCT01451853, phase 2) was designed to investigate the safety and efficacy of this compound in preventing hearing loss induced by platinum-based cytostatics (e.g., cisplatin, carboplatin). Ebselen was given orally to participants twice daily for three days of each chemotherapy cycle in three different dose schemes (low: 200 mg, middle: 400 mg, and high: 600 mg) depending on the trial arm [65].

Another compound from the 1,2-benzisoselenazole-3[2H]-one derivatives class is **ethaselen** (1,2-[bis(1,2-benzisoselenazolone-3(2H)-ketone)]ethane, 2-[2-(3-oxo-1,2-benzoselenazol-2-yl)ethyl]-1,2-benzoselenazol-3-one, BBSKE, Figure 5, structure 18), which seems to be a very promising molecule with potential anticancer properties [6]. This compound has been identified as a mixed-type mammalian TrxR inhibitor and its goal is the C-terminal active site of this enzyme. By binding to the redox pair Sec-Cys, it inhibits the reduction of oxidized Trx, leading to the accumulation of both ethaselen and ROS in the cell [8,66]. The suppressive effect of ethaselen on the proliferation of cancer cells has been confirmed on many cell lines in vitro and in vivo, including cancer models of the lung, tongue, stomach, liver, colon, prostate, cervix, nasopharyngeal cavity, and leukemia [7,8,66,67].

As far as the mechanism of action and the properties of this compound at the chemical level are concerned, it turns out that the place with strong electrophilicity is the bond between the selenium and nitrogen (Se-N) atoms in the benzisoselenazole ring, which affects the ability to react with -SH and -SeH groups of proteins. As a result of this reaction, diselenide (Se-Se) and selenenylsulfide (Se-S) bonds are formed between ethaselen and TrxR molecules [66]. In their work, Wang et al. [66] used the molecular docking simulation of ethaselen in the TrxR enzyme. Depending on the TrxR type, only S-Se or S-Se and Se-Se bonds could be formed. In the first stage, the nucleophilic Cys498 (Sec498 in TrxR wild type) attacks the S-N bond in the benzisoselenazole ring of ethaselen, which leads to its opening and the formation of the S-Se bond (Se-Se in the case of the wild type enzyme). In the second stage, the opening of one of the two rings increases the elasticity of the compound and causes the interaction of the S-N bond in the second ethaselene ring with Cys497 and the formation of the S-Se bond between it and the enzyme. Apart from this, the authors suggest that hydrogen bonds between His472 and Tyr116 residues and ethaselen are also involved in this mechanism, so it could affect the catalytic functions of TrxR1. As mentioned earlier, the inhibition of TrxR caused the accumulation of oxidized Trx, which resulted in a decrease in NF-κB signaling and the introduction of cells into the apoptosis pathway [67]. In addition, in Tca8113 tongue cancer cells, it was observed that as a result of TrxR inhibition, caspase 3 was activated [7].

In addition to effective ethaselen monotherapy, there is evidence that this compound increased efficacy in combination with available cytostatic drugs or radiotherapy. This compound enhanced the effectiveness of cisplatin in human lung xenografts (A549) in mice [8,67] and leukemic cells resistant to cisplatin, and acted synergistically with sunitinib against colon cancer cells [8]. In addition, it sensitized to radiotherapy non-small cell lung cancer cells (NSCLC) [12,66].

Currently, this compound is in phase 1c clinical trials (NCT number: NCT02166242) in China, where patients with non-small cell lung cancer (NSCLC) take a dose of 600 mg of ethaselen/day (safe and tolerable dose: 1200 mg/day as determined in phase 1a/b) [68].

### 3.4. Selenophene-Based Derivative

Among this class of compounds, great curiosity is aroused by **2,5-bis(5-hydroxymethyl-2-selenienyl)-3-hydroxymethyl-N-methylpyrrole** (D-501036, Figure 6), which has a wide range of activities against some human cancer cells lines. Its effect depends on the used dose and exposure time. It was demonstrated that exposure to D-501036 caused an increase in caspase 3 and 9 activity [7]. In addition, this compound was selective in relation to cancer cells, causing apoptotic events and DNA double-strand breaks (DSBs). Its high effectiveness was observed in relation to cells exhibiting multidrug resistance (MDR) characterized by the overexpression of glycoprotein P, which gives hope for an increase in treatment efficacy in chemotherapy-resistant cancers [6].

### 3.5. Seleninic Acids

One of the most extensively investigated Se-containing compounds with anticancer properties, apart from selenite, is **methylseleninic acid** (methylselenic acid, methaneseleninic acid, MSA, Figure 7). It is a compound from the group of oxoacids [8], which is obtained from methylselenocysteine (MSC) during the transamination reaction or with the involvement of β-lyase, i.e., this enzyme is not necessary for MSA activity [3,6,69].

In vitro, MSA has a stronger activity than MSC, but in vivo, this difference is minimized (β-lyase is present in tissues) [3]. MSA is non-genotoxic in comparison with selenite [8,27] and does not exhibit toxicity to normal body tissues [8,10,70,71]. Its impact on cancer cells was studied both in vitro and in vivo on various cell lines, i.e., prostate [24,25,26,27,69,72,73,74], colon [10], breast [32,75], liver [34], lung [33], and pancreatic [8] cancer. MSA inhibits the proliferation of cancer cells by inducing an apoptotic process via various mechanisms [8,30,34], arresting the cell cycle in phase G1 [3,26,27] and its antiangiogenic activity [10,27,30,75]. One of the possible mechanisms of MSA is the excessive generation of ROS [33].

In a study on human hepatoma HepG2 cells, it was noted that MSA (25 µM) was a strong oxidant that led to a rapid depletion of intracellular glutathione (GSH) as a result of the reaction with it and the formation of anticancer-active methylselenol [34]. It can therefore be predicted that cancer cells with increased levels of GSH in a cell will be more sensitive to the effects of MSA. As it is generally known, excessive levels of ROS in a cell cause ER stress and the induction of UPR [8]. Moreover, an important element is caspase 12 (present in ER) and its activation during ER stress, which triggers an apoptotic process in PC-3 cells [30]. The mechanism in which ER and UPR are involved was described in a study conducted by Shigemi et al. [37] using primary effusion lymphoma (PEL) cells, in which treatment with MSA increased the levels of oxidized proteins and promoted ER stress, inducing proapoptotic UPR and then leading the cell to the apoptotic pathway. In relation to the above, these results suggest that apoptosis in cancer cells may occur through a disturbed cell redox balance [30].

Meanwhile, in another study, apoptosis was ROS-independent [24]. MSA treatment of prostate cancer DU145 and PC-3 androgen-independent cells caused DNA fragmentation and caspase-dependent PARP cleavage, which indicates caspase-dependent apoptosis [5,26]. It is worth adding that apoptosis was caspase-dependent (caspase 3, 7, 8, 9), but it was not affected by p53 [8,10,24]. In a study conducted by Gasparian et al. [25], a different mechanism of anticancer action of MSA was observed. Prostate cells treated with 5 µM MSA were directed to the apoptotic pathway as a result of the inhibition of NF-κB DNA binding stimulated by TNF-α and lipopolysaccharide (LPS). This resulted from blocking the activation of the IκB kinase (IKK) by MSA and preventing the release of NF-κB. At this point, it is worth mentioning that interleukin 6 (IL-6) activates NF-κB, enabling its binding to DNA [76], because in the study carried out by Zeng et al. [10], an inhibition of proliferation in colon cancer xenografts in C57BL/6 mice as a result of treatment with MSA (3 mg/kg body weight) and a corresponding decrease in TNF-α and IL-6 levels were observed. Therefore, in this situation, it seems that NF-κB DNA binding could also be inhibited.

Summarizing these two investigations, it can be suggested that the inhibition of the inflammatory process induced by LPS, TNF-α, and IL-6 may have an impact as an anticancer and chemopreventive factor. Moreover, other MSA mechanisms of directing cancer cells to the apoptotic pathway were observed, such as the inactivation of PKC [73], inhibition of HDAC [6,32], blocking of androgen [71,74] or estrogen receptor (ER) [5] signaling, upregulation of CDK inhibitors (CDKI), which are inhibitors of kinases CDK2, -6 and -4 [7], and other molecular targets, such as the proteins regulated in development and DNA damage response 1 (REDD1), ERK 1/2, p38MAPK, JNK 1/2 or protein kinase AKT [3,26,69]. As mentioned earlier, MSA exhibits angiogenesis inhibitory properties. In investigations on angiogenesis inhibition, downregulation of the expression and levels of hypoxia-inducible factor 1α (HIF-1a), vascular endothelial growth factor (VEGF), matrix metalloproteinase-2 (MMP-2), angiopoietin-2 (Ang-2) and integrin β3 (ITGB3, CD61) was observed [8,26,72,75]. In turn, disorganization of CD61 aggregates interrupted the phosphorylation of AKT, IκBα, and NF-κB [8]. Apart from apoptosis, there are also other types of cell death. Interestingly, it was observed that MSA induced necrosis in MCF-7 [32] and U-2 OS [36] cells.

In the course of the research, it was also found that MSA enhances the effectiveness of several chemotherapy drugs, i.e., paclitaxel [77,78], TAM [70], doxorubicin, cytosine arabinoside [6], taxol, etoposide, 7-ethyl-10-hydroxycamptotecin (SN-38) [77], and cyclophosphamide [79]. The applied combination of MSA (4.5 mg/kg b.w./day) and paclitaxel (microtubule inhibitor; 10 mg/k b.w./day) in the xenografts of triple-negative breast cancer in SCID mice showed synergistic enhancement of the therapeutic effect of paclitaxel, which was the result of arresting the cell cycle at the G2/M checkpoint and a significantly higher percentage of apoptotic cells in which the induction of apoptosis occurred through caspases [78]. In a study by Hu et al. [77], MSA amplified the apoptotic effects of paclitaxel (10 µM), etoposide (topoisomerase II inhibitor; 15 µM), and SN-38 (topoisomerase I inhibitor) in prostate cancer DU145 cells by affecting JNK-dependent molecular targets, amplifying the cascade of caspase 8 activity. Additionally, it was found that MSA can intensify cell uptake/retention of SN-38.

In turn, Li et al. [70] concluded that combination therapy of TAM + MSA against TAM-resistant and TAM-sensitive breast cancer cell lines increased the effectiveness of TAM and sensitization of TAM-resistant cells. The sensitization mechanism consisted of arresting the cell cycle in phase G1 by TAM, which permitted more cells to enter the mitochondrial apoptotic pathway induced by MSA. Another possible mechanism for MSA + TAM is a loss of ERα signalization. Among other drugs, MSA in combination with ABT-737 (inhibitor of Bcl-2 family proteins) increased caspase 8, 9, and 3 activity, introducing more cells into the apoptotic pathway in PC-3, MDA-MB-231, and HT-29 cell lines [7].

Additionally, contrary to selenite, MSA synergistically enhanced the proapoptotic properties of tumor necrosis factor-related apoptosis-inducing ligand (TRAIL) in DU145 prostate cancer cells, which was associated with the internal mitochondrial pathway [30]. Except for the synergistic enhancing efficacy of commonly used cytostatic agents by MSA, Lafin et al. [80] recently found that this compound also sensitizes head and neck squamous cell carcinoma (HNSCC) to radiotherapy. The mechanism involved in this phenomenon was the induction of lipid peroxidation (LPO) by MSA. Very importantly, MSA was non-toxic to normal skin cells. These results suggest that MSA may be an effective adjuvant during radiotherapy in HNSCC. In conclusion, MSA has very promising anticancer potential both alone and in combination with other cytotoxic drugs and radiation.

### 3.6. Selenoesters

The design and synthesis of alkyl and aryl selenoesters were guided by the creation of compounds that would be enzymatically reduced (may be TrxR1 substrates [81]) or hydrolyzed (mediated by a nucleophilic agent such as water [15] or enzymes such ascarbonic anhydrases [82] or acetylcholinesterase [83]) to metabolites (i.e., selenol, methylselenol) exhibiting redox activity in cells. These small active molecules can disrupt redox processes in cells, finally leading to cancer cell death [8]. The cytotoxicity of these derivatives has been studied on colon, prostate, breast, lung, and T-lymphoma cell lines, showing activity even at nanomolar concentrations [8,84,85]. **Selenoesters with ketone end fragments** showed the highest activity (Figure 8, structures 19–21). Moreover, derivatives with ketone terminal fragments inhibited the efflux pump ABCB1, responsible for MDR, more strongly than the reference compound verapamil. In mouse MDR T-lymphoma cells, the strongest inhibitor of ABCB1 (glycoprotein P, P-gp) was **Se-2-oxopropyl 4-chlorobenzoselenoate** (Figure 8, structure 19), which was 3.4-fold more potent than verapamil [85].

Similarly, in the case of MDR Colo 320 human adenocarcinoma cells, the same compound **19** exhibited a four-fold greater inhibition of P-gp than the reference compound and was the most selective of the most active compounds **19–21** in this study [39]. It is worth noting that in both studies, the concentration of compound **19** was 10-fold lower than verapamil (2 µM **19**, 20 µM verapamil) [39,85]. All compounds **19–21** induced apoptosis in mouse MDR T-lymphoma cells, wherein compounds **20** and **21** induced early and compound **19** induced late apoptosis [85]. In addition, these compounds **19–21** seem to be interesting agents for use as adjuvants in chemotherapy. In the experiments carried out by Spengler et al. [86], it was observed that compounds **19–21** exerted synergistic interactions with topotecan and vincristine, whereas additionally compounds **19** and **20** interacted with doxorubicin (Dox) and compound **19** interacted with cyclophosphamide. This would suggest that these compounds could affect the formation of microtubules and cell topoisomerases. However, it is surprising that these compounds **19–21** interacted antagonistically with verapamil, which, in comparison with the results of the two previous studies, where inhibition of efflux pump ABCB1 was shown, would suggest possible competition for a transporter between compounds **19–21** and verapamil.

Compounds **22** (methyl 3-chlorothiophen-2-carboselenoate, Figure 8) and 23 (dimethyl 2,5-furandicarboselenoate, Figure 8) were tested on cancer cell lines of the pancreas (PANC-1), lung (HTB-54), breast (MCF-7), prostate (PC-3), colorectal (HT-29), and chronic myeloid leukemia (K-562) [15,81]. Díaz-Argelich et al. [81] found that cell proliferation was inhibited in micromolar concentrations <10 µM (except for K-562 cells, where IC_50_ was 38.7 and 42 µM for **22** and **23**, respectively) with cell cycle arrest in the G2/M phase. Growth inhibition was induced by these agents in a dose-dependent manner, whereas the effects of compound **23** were additionally time-dependent. The investigators’ assumption was the synthesis of compounds which, after hydrolysis, would release methylselenol, a key metabolite with anticancer properties. However, during the study, it turned out that compounds **22** and **23** are TrxR1 substrates and a reduction through this enzyme is more effective than hydrolysis, which could increase methylselenol generation. In addition, it was confirmed that they did not exhibit free radical scavenging properties and that cell death partly depended on the caspase pathway.

Entosis is a form of cancer cell cannibalism, in which a living cell is absorbed into the cytoplasm of another cell and then digested by lysosomal enzymes. This process is based on the principle of autophagy but without the presence of autophagosomes [87]. In another study carried out by Khalkar et al. [15], it was observed that derivatives of methylselenoesters **22** and **23** caused a drop in cell adhesion to the surface as a result of a decrease in the expression of cell division control protein 42 homolog (CD42) and a discontinuation of integrin β1 (CD29) signaling. The detachment of cells and a simultaneous increase in N-cadherin levels caused the cells to clump into grape-like aggregates, which led to the absorption of one cell by another (entosis) and finally the death of the trapped cell.

The compounds **24** (dimethyl benzene-1,4-dicarboselenoate, Figure 8) and **25** (dimethyl pyridine-2,6-dicarboselenoate, Figure 8), synthesized by Domínguez-Álvarez et al. [84], were tested for their cytotoxicity. Treatment with compounds **24** and **25** in nanomolar concentrations inhibited the growth of PC-3 prostate cancer cells. Furthermore, derivatives **24** and **25** were the most selective among all the compounds in the panel, as well as the reference drugs (etoposide, cisplatin), and did not exhibit antioxidant properties, the same as **19**. In addition, both compounds interacted synergistically with vincristine, cyclophosphamide, Dox, and methotrexate (MTX) in concentrations higher than **19–21** and showed moderate synergy with verapamil [86]. Regarding their effect on MDR, **24** and **25** marginally affected the activity of the P-gp efflux pump; their effect at both concentrations (2 and 20 µM) was lower than that of verapamil in both mouse MDR T-lymphoma cells [85] and MDR Colo 320 human adenocarcinoma cells [39].

### 3.7. Selol

**Selol** is a very interesting compound containing in its structure selenium at +4 oxidation stage [88]. It was obtained at the Medical University of Warsaw as a result of a reaction of triglycerides from sunflower oil with selenic acid (IV) [89,90]. The study, conducted by Rahden-Staroń [88], showed that this compound is non-toxic and non-mutagenic. However, its toxicity to the body depends on the route of administration. Selol administered parenterally was non-toxic, whereas during oral administration its toxicity increased sharply, which may indicate the formation of more harmful products during digestion [89]. For this reason, it should be considered to be administered only parenterally. Moreover, the activity of Selol is determined by its Se content—the number of dioxaselenolane rings (Selol 2%—single rings, 5%> single and double rings, Figure 9). The authors found that 2% Selol exerted antioxidant activity through the induction of enzymes of the second phase of detoxification, which would prove its chemopreventive properties, while 7% Selol exhibited cytotoxic properties on human colorectal adenocarcinoma Caco-2 cells [91].

Additionally, it is worth noting that its antioxidant activity and high bioavailability result from the incorporation of Se from Selol into selenoproteins [92]. Selol was tested on various cancer cell lines, including prostate cancer and leukemia [93,94]. In vitro experiments on HL-60 and vicristine/doxorubicin-resistant (HL-60/Vinc and HL-60/Dox) leukemia cells showed that Selol inhibited cell proliferation and induced apoptosis, affecting resistant lines more strongly [93]. In the androgen-dependent prostate cancer (LNCaP) cell line, this compound inhibited cell proliferation and probably directed them to the apoptotic pathway, not showing this activity against normal cells [94]. Additionally, 5% Selol is a prooxidant (causes excessive ROS production) [95] and was also recognized as a TrxR inhibitor by Sochacka et al. [96].

In research on the interactions of this compound with the commonly available drugs used in chemotherapy, it was reported that it enhanced the antiproliferative effects of Dox, especially in cells that were resistant to this drug [97]. Most importantly, in vincristine-induced hyperalgesia, Selol enhanced the analgesic effects of fentanyl, buprenorphine, and morphine, which gives hope for a new therapy in palliative care in terminal states of cancers [98]. Despite the promising results, the potential molecular targets and the detailed mechanism of Selol’s activity need to be thoroughly investigated in the future.

## 4. Selenoamino Acids

Among the Se-containing amino acids, which exhibited anticancer activity in studies, we can distinguish a.o. compounds whose formulas are included in Figure 10. Selenocystine (**29**) is a naturally occurring amino acid [99], while selenomethionine (**26**), methylselenocysteine (**27**)**,** and selenocysteine (**28**) are contained in Se-enriched yeast [4]. Most importantly, these amino acids are better absorbed, metabolized, and stored by the human body [6]. These selenoamino acids are incorporated into selenoproteins in the form of selenocysteine, e.g., Gpxs and TrxRs, or metabolized into low molecular weight compounds [100]. In the following subsections, each of these substances and their activities against cancers will be discussed.

### 4.1. Selenomethionine

**Selenomethionine** (l-selenomethionine, seleno-l-methionine, SeMet, Figure 10, structure 26) is the major selenoamino acid contained in Se-enriched yeasts [3], which has very low toxicity and is non-genotoxic [4,27,101]. Studies have indicated that the tissue accumulation of selenium was better when SeMet was ingested with the diet compared to other forms of selenium [5,27,101]. This may be due to the fact that SeMet is unspecifically incorporated into proteins instead of methionine (Met), so chronic supplementation of SeMet may be risky, because during high tissue breakdown of the body, the occurrence of selenosis is possible [8,27,29]. As SeMet is not a redox-active compound [3,28,29], the l-methionine-α-deamino-γ-mercaptomethane lyase (l-methioninase, METase, γ-lyase) enzyme is necessary for the formation of the active metabolite (methylselenol) [3,30,100,102]. Inhibition of cell proliferation due to apoptosis induction caused by SeMet was observed in several cancer cell lines, i.e., colorectal, prostate, breast, lung, and melanoma [3,8].

For prostate cancer cells, SeMet inhibited proliferation and directed the cells to the apoptotic cell death pathway [5]. In p53-positive lung cancer cells A549, the percentage of apoptotic cells increased after SeMet treatment [12]. There was no co-existing autophagy during apoptosis [7], but there was a reduction in polyamine levels in lung cancer cells [3,5]. In HT-29 cancer cells, there was also a drop in polyamine concentration [5]; while in HCT116 cells, changes in mitotic cyclin expression and stopped activation of cdc2 kinase were observed [3]. Since a relationship between the high expression of COX-2 and an increase in the progression and avoidance of apoptosis in colorectal cancer cells was observed and described [103], it would be a rational explanation that a decrease in COX-2 levels at least partially inhibits the proliferation of these cells. A confirmation of this thesis is provided by studies in which SeMet treatment resulted in the inhibition of cancer cell growth, which correlated with a fall in COX-2 levels [3,5]. This process may also be affected by a decline in polyamine levels, which takes part in the post-transcriptional regulation of COX-2 [3], but this requires further, more detailed research. In regards to the effect of SeMet on p53 protein and apoptosis, it was found that the apoptotic process in A549 cancer cells is p53-dependent [7]. In addition, it was observed that at low doses (in the range of 10–20 µM SeMet), there was a modulation of p53, which was Ref1-dependent, resulting in the occurrence of a p53 form that stimulated DNA repair without affecting proliferation [3,100].

As mentioned above, SeMet is degraded by METase into an active metabolite, methylselenol. In studies on the cancer cells HT1080, HCT116, and DU145, methylselenol was generated topically (in situ) from SeMet using METase, and it was observed that the ERK 1/2 pathway was blocked and cells were arrested in phase G1 of the cell cycle [3,11,104]. Apart from the abovementioned mechanisms of introducing cells to the apoptosis pathway, it was reported that SeMet can also induce this process by inhibiting Gpx [8] and HDAC [105], ER stress [8], activating p53 and caspases [3], and changes in the expression of anti-apoptotic (Bcl-xL) and proapoptotic (Bax, Bim, Bad) genes [8]. Inhibition of proliferation and processes associated with apoptosis are dose- and cancer cell line-dependent. In a review carried out by Sinha and El-Bayoumy [3], they concluded that the cytotoxic effects of SeMet on cancer cells appeared at concentrations of 45–150 µM, while in normal fibroblast cells, inhibition of proliferation required 1 mM SeMet. The non-toxicity of SeMet in relation to normal cells was also confirmed by Shen et al. [28], who estimated that this compound did not cause inhibition of NHK cell growth even at a concentration of 316.6 µM Se. However, a recent study from 2017 [9] showed that the cytotoxicity of SeMet to human glioblastoma multiforme cells GMS-10 and DBTRG-05MG was induced by significantly higher doses in the range of 500 and 1000 µM, while at doses of 50 and 100 µM SeMet, cell growth was stimulated. There are also reports that SeMet prevents B16BL6 cells from metastasizing into the lungs [3] and that it also increases the selectivity of radiotherapy to NCI-H460 and H1299 cancer cells with a slight impact on normal cells [8]. It should not be forgotten that SeMet has been used in many clinical trials that concerned its chemopreventive effect, improving the effectiveness or reducing the side effects of commonly used treatments, as shown in Table 2.

### 4.2. Methylselenocysteine

Another naturally occurring selenoamino acid is **methylselenocysteine** (Se-Methyl-selenocysteine, Se-methylseleno-l-cysteine, MSC, Figure 10, structure 27), which is also not a redox-active compound like SeMet [8]. Its activity is determined by the effective metabolism to methylselenol by β-lyases present in mammalian cells [3,5,8,13,121]. In addition, this compound is characterized by high bioavailability [13] and is non-genotoxic, non-toxic, and well tolerated by the organism [13,27,101]. Its cytotoxicity and ability to induce apoptotic events at micromolar concentrations was proven in several models of cancer cell lines in vitro and in vivo, such as prostate, breast, ovarian, colon, oral squamous cells cancer, osteosarcoma, and promyelocytic leukemia [3,4,8,12,13,27,36]. In in vitro studies, it was observed that in HSC-3 cells with mutant p53, there was an increase in apoptosis after MSC treatment [12]. Similar observations were reported in a model of osteosarcoma, where the apoptotic process was confirmed by changes in cell morphology and their ultrastructure [36]. In addition, IC_50_ values were established after 72 h of MSC treatment for lung A549 (100 µM) and hepatoma HepG2 (177 µM) cancer cells [13].

Moreover, in a study using human LNCaP human prostate cancer xenografts in mice, it was reported that the growth inhibition of tumors was accompanied by the downregulation of the androgen receptor (AR) and a decrease in prostate-specific antigen (PSA) levels [121]. Studies using DU145 prostate cancer xenografts in athymic nude mice [27] and transgenic prostate adenocarcinoma (TRAMP) [71] models showed that MSC at a dose of 3 mg Se/kg b.w. inhibited tumor growth and was well-tolerated. Treatment with MSC in the TRAMP model was associated with increased survival and reduced metastases to other organs [71]. Previous papers found that MSC treatment resulted in cell cycle arrest in phase G1 [27,36] or early phase S [4,5,122], and apoptotic events were caspase-dependent (↑ activity of caspase 3, 8 and 9, which leads to PARP cleavage) [3,7,8,12,27]. One of the molecular targets of MSC is also PKC, which was specifically blocked when cancer cells were treated using this compound. The consequence of this event was a reduction in cdk2 kinase activity and DNA synthesis, as well as an intensification of gadd gene expression, which led to the introduction of cells into the apoptotic pathway [29,122]. Moreover, it could be suggested that apoptosis could occur via the internal pathway (mitochondrial-dependent), because a rise in proapoptotic proteins (Bax, Bad) with a simultaneous drop in anti-apoptotic proteins (Bcl-2, Bcl-xL) was observed [12,36].

Another important anticancer mechanism of MSC is the inhibition of tumor angiogenesis [4,6,8,13,27,121]. At the basis of this process is a decline in the expression of proangiogenic factors such as VEGF, HIF-1α, COX-2, and iNOS [8,13]. Additionally, MSC promoted an increase in blood vessel maturation [6,13]. Other mechanisms involved in the antiproliferative activity of MSC include HDAC inhibition [105], ROS generation, and an impact on molecules such as PI3K, Akt, p38MAPK, survivin, HIAP1, and XIAP [3,7].

Furthermore, the indication of the synergistic action of MSC with commonly used cytostatic drugs was very important evidence. Combined therapy with MSA and TAM in MCF-7 breast cancer xenografts in female mice resulted in synergistic inhibition of tumor development. This was caused by an elevated percentage of apoptotic cells, decreased levels of ERα protein and its lower signaling, as well as its antiangiogenic activity [121]. A synergism of combined treatment was also observed when MSC was used together with irinotecan [7,8], which may be caused by a greater concentration of the active metabolite irinotecan (SN-38) in the cell. Moreover, an improvement in efficacy was observed in sensitive (FaDu, HCT-8) as well as drug-resistant (A253, HT-29) tumors [8]. The inhibition of angiogenesis by MSC synergistically improved the effectiveness of treatment with Dox in xenografts of FaDu cells in mice, leading to greater transport of Dox to tumor cells and consequently the inhibition of proliferation. The improved delivery of Dox resulted from functional and normal (mature) blood vessels and a reduction in their permeability and density [8]. An additional benefit of combining MSC with other cytostats (cisplatin, oxaliplatin, the above-mentioned compounds, and others) is a reduction in the toxicity of these drugs [6], e.g., the prevention of acute myelosuppression in Fisher rats induced by oxaliplatin [13]. There are also human trials on chemoprevention and reduction in side effects of chemotherapy by MSC, as summarized in Table 3.

### 4.3. Selenocysteine

A very important selenoamino acid for humans is **selenocysteine** (l-selenocysteine, Sec, Figure 10, structure 28), which is essential for the proper functioning of the body, because its residues are present in active centers of enzymes showing antioxidant properties [125,126]. The physiological role of Sec was widely reviewed in Ref. [127]. Moreover, it is non-toxic to the organism [36] and is a component contained in Se-enriched yeasts [4]. Sec inhibits the growth of cancer cells by inducing apoptosis and arrest of the cell cycle in phase S [36,128]. It was observed that downregulated cyclin A and cyclin-dependent kinase-2 (CDK-2) were linked with cell cycle arrest. ROS generation and p53 modulation also participate in the anticancer activity of Sec [7,36]. ROS generated by Sec caused the oxidation of protein thiol groups in a HepG2 hepatoma cancer model [30].

As far as the p53 protein is concerned, Sec activates p53 early as a result of phosphorylation mainly of the residue Ser 15 p53 [12,36]. Then, the activation of p53 induces a rise in proapoptotic (Bax and Bad) and a decline in antiapoptotic (Bcl-2, Bcl-xL) protein expression [36,99], which in turn results in a loss of mitochondrial membrane potential (ΔΨm) [12,36,99]. Dysfunctional mitochondria release cytochrome c and AIF, which finally leads to DNA fragmentation [12,99]. Additionally, apoptosis occurs via a caspase-dependent pathway [12,128]. This happens as a result of caspase 9 activations via ↑ Bax/Bad and cytochrome c in a cell, which then activates caspase 3 and 7 (executioner caspases), causing PARP cleavage [36,99]. To summarize the above mechanisms, the internal pathway, cascade of caspases, ROS generation, and modulation of p53 protein take part in the apoptosis induced by Sec. In in vitro and in vivo studies on human melanoma cells, Sec induced caspase-dependent apoptosis [128].

In studies with the use of combined therapy, it was reported that Sec and 5-FU had a synergistic effect [12]. The same effect was observed in the case of Sec and auranofin treatment in breast [128] and lung [85] cancer cell lines. Apart from that, this compound sensitized cervical cancer cells to radiotherapy, resulting from the generation of excessive ROS levels in cells [12].

In addition, selenocysteine derivatives, such as **3,3′-diselenodipropionic acid** (DSePA) and **Se-alliloselenocysteine** (ASC), can have both protective and anticancer effects. ASC induced α and π isoforms of glutathione-S-transferase, thus protecting cells from carcinogenesis [47], whereas in human colorectal cancer cells HT-29 and HCT-116, this compound inhibited cell growth and induced autophagy through the AMPK/mTOR pathway [129]. Meanwhile, in the case of DSePA, it was found that this compound had a protective effect on albino mice exposed to γ-radiation [7]. Apart from that, DSeAP synergistically enhanced the proapoptotic activity of TRAIL in melanoma cells (A375), indicating DSeAP as a potential candidate with chemosensitizing properties. The generation of excessive amounts of ROS after DSeAP treatment finally led to the phosphorylation of p53 protein and breakthrough of resistance of the PI3K/Akt-mediated pathway [128].

### 4.4. Selenocystine

Looking at the structural formula of **selenocystine** (SeCys, Figure 10, structure 29), it can be seen that it is a diselenide dimer of the previously described selenoamino acid, **selenocysteine** (Sec, Figure 10, structure 28). SeCys is generated as a result of SeC oxidation [8,130]. This compound is characterized by a dualism of activity—it may exhibit both antioxidant as well as prooxidant properties [8]. It is suggested that its antioxidant activity is achieved at nutritional levels [13]. However, SeCys is a compound that exhibits redox activity only after reduction to Sec, which is more nucleophilic than SeCys and thus less stable and more reactive, and in the reduction reaction disulfide reductases or low molecular weight thiols are involved [8,13]. Investigations on cancer cell models of melanoma, cervical, breast, liver, or lung cancer indicated that Sec inhibits cell growth in micromolar (low to medium) concentrations; however, the mechanism of its action and the method of the initiation of cell death depend on the cancer cell line [8,130].

As a diselenide dimer of Sec, SeCys exhibits a similar mechanism of cytotoxic activity to Sec, which includes the generation of ROS [130] and apoptosis dependent on the internal (mitochondrial) pathway and p53 protein modulation (as described above in Sec) [131]. In addition, SeCys inhibits PKC activity [29] and can direct cells to the parapoptotic pathway of cell death [8,13]. The occurrence of apoptotic or parapoptotic cell death depends on the phase of the cell cycle in which the cell is actually found [8]. The parapototic pathway is associated with ER stress, stimulation of UPR, and a high percentage of vacuoles in the cell cytoplasm [13]. In normal human fibroblasts Hs68, IC_50_ > 400 µM and no ROS generation was observed, contrary to the detection of these free radicals in cancer cells (A375, HepG2, MCF-7) and IC_50_ in the range of 3.6–37 µM, which would mean that SeCys only affects cancer cells [130].

Similar observations were described in a mouse xenograft melanoma model, where no toxic effect of SeCys on the mouse body was found [8]. A study conducted by Weisberger and Suhrland [132] on people suffering from acute or chronic myeloid leukemia gave very interesting results concerning SeCys and its activity. SeCys treatment exhibited significant efficacy against immature leukocytes compared to mature leukocytes and simultaneously did not have an impact on the bone marrow (was non-toxic). Additionally, in the same study, during SeCys treatment, sensitization to the available cytostatic agents was reported in patients who were resistant to them. These results suggest that the application of SeCys may be a therapeutic alternative to anticancer treatment, especially leukemia.

Alongside this, there are reports of the synergistic enhancement of the available drugs used in chemotherapy by SeCys. An improvement in the therapeutic response in Dox treatment was observed in combination with SeCys in HepG2 hepatoma cancer cells. In turn, a combination of SeCys and auronafin (TrxR inhibitor) potentiated the effects of this cytostatic and increased the proportion of apoptotic cells in lung cancer cells A549 [39]. This was caused by excessive levels of generated ROS, leading to TrxR knockdown [8].

## 5. Conclusions

Year by year, more and more incidences of cancer are being reported. The increasing resistance of tumors to standard chemotherapy or radiotherapy, as well as the severe side effects associated with them, are prompting the search for new anticancer agents. The properties of selenium compounds give new hope in this area, as shown by the results of many of the studies that were described in this review. The reasons for using selenium compounds as new potential agents or co-adjuvants in cancer treatment are less toxicity, greater selectivity and effectiveness, as well as reducing the adverse effects of commonly used anticancer therapies. In conclusion, despite the potential of these Se-containing compounds, further extensive research is needed, mainly in vivo, to determine the impact of their long-term intake and the distant effects associated with their use.

## Figures and Tables

**Figure 1 ijms-22-01009-f001:**
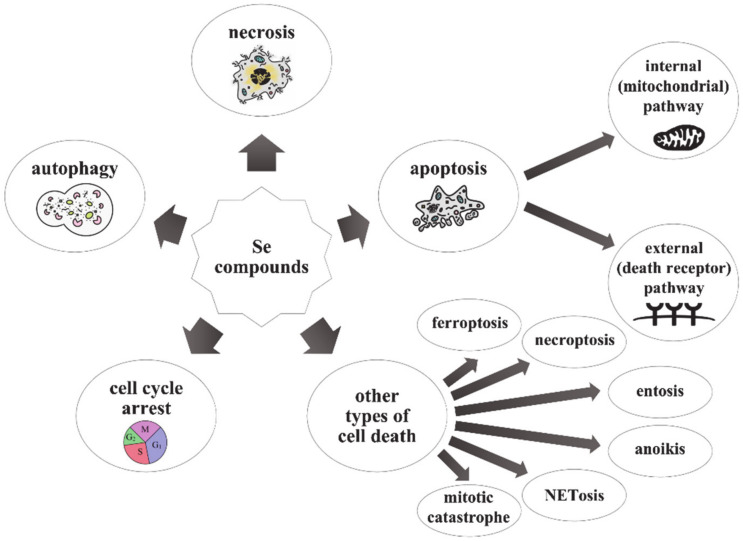
Types of cell death induced by Se-containing compounds.

**Figure 2 ijms-22-01009-f002:**
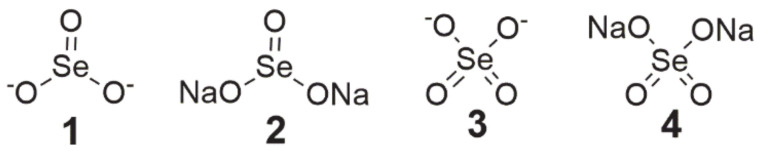
Chemical structure of inorganic selenium compounds (**1**—selenite, **2**—sodium selenite, **3**—selenate, and **4**—sodium selenate).

**Figure 3 ijms-22-01009-f003:**
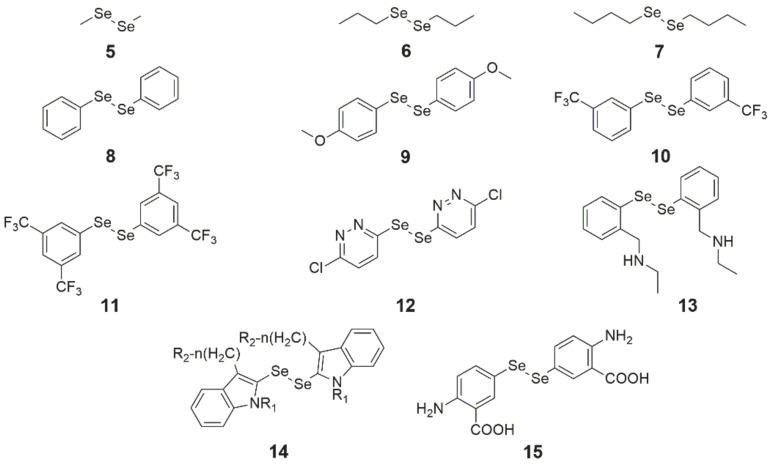
Chemical structures of diselenide derivatives (**5**—dimethyl diselenide, **6**—dipropyl diselenide, **7**—dibutyl diselenide, **8**—diphenyl diselenide, **9**—4,4′-dimethoxy-diphenyl diselenide, **10**—3,3′-ditrifluoromethyl-diphenyl diselenide, **11**—3′,5′,3,5-tetratrifluoromethyl-diphenyl diselenide, **12**—1,2-bis(chloropyridazinyl) diselenide, **13**—N,N’-((diselanediylbis(2,1-phenylene))bis(methylene))di(ethaneamine), **14**—indol-containing diselenide, **15**—bis(4-amino-3-carboxyphenyl) diselenide).

**Figure 4 ijms-22-01009-f004:**
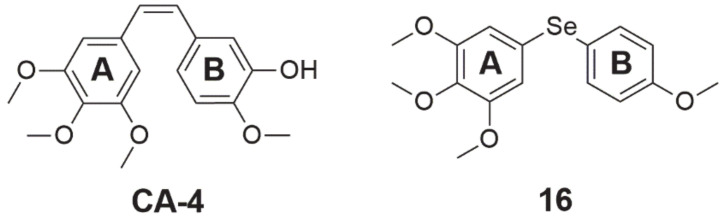
Chemical structures of combretastatin 4-A (**CA-4**) and its derivative (**16**).

**Figure 5 ijms-22-01009-f005:**
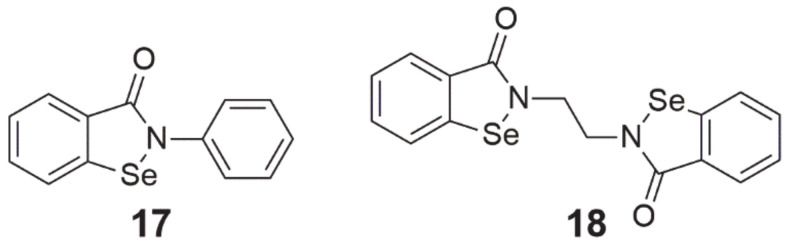
Chemical structures of 1,2-benzoselenazole-3[2H]-one derivatives (**17**—ebselen, **18**—ethaselen).

**Figure 6 ijms-22-01009-f006:**
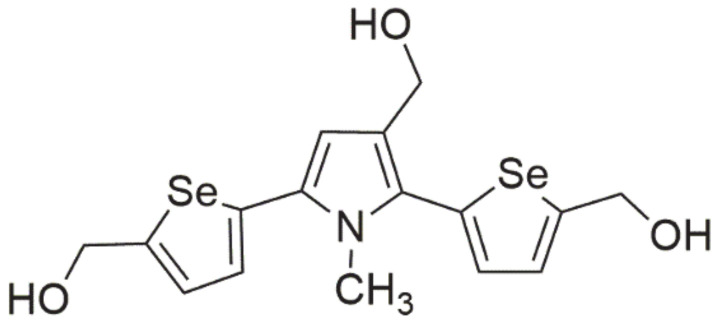
Selenophene-based derivative (D-501036).

**Figure 7 ijms-22-01009-f007:**
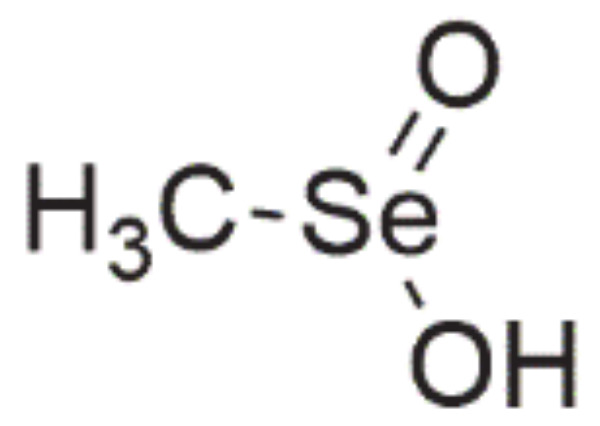
Chemical structure of methylseleninic acid.

**Figure 8 ijms-22-01009-f008:**
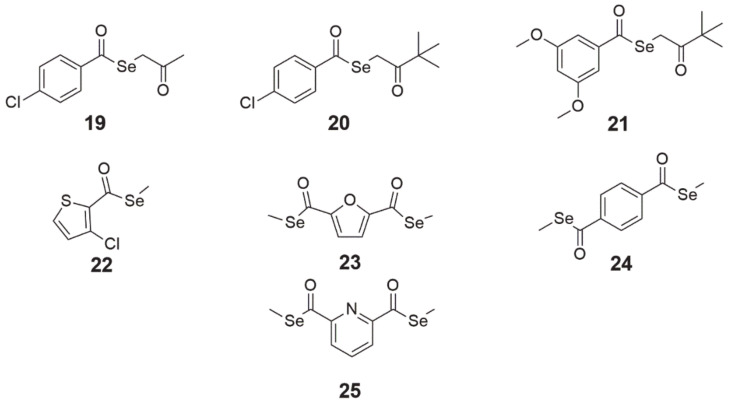
Chemical structures of selenoester derivatives (**19**—Se-2-oxopropyl 4-chlorobenzoselenoate, **20**—3,3-dimethyl-2-oxobutyl 4-chlorobenzoselenoate, **21**—3,3-dimethyl-2-oxobutyl 3,5-dimethoxybenzoselenoate, **22**—methyl 3-chlorothiophen-2-carboselenoate, **23**—dimethyl 2,5-furandicarboselenoate, **24**—dimethyl benzene-1,4-dicarboselenoate, **25**—dimethyl pyridine-2,6-dicarboselenoate).

**Figure 9 ijms-22-01009-f009:**
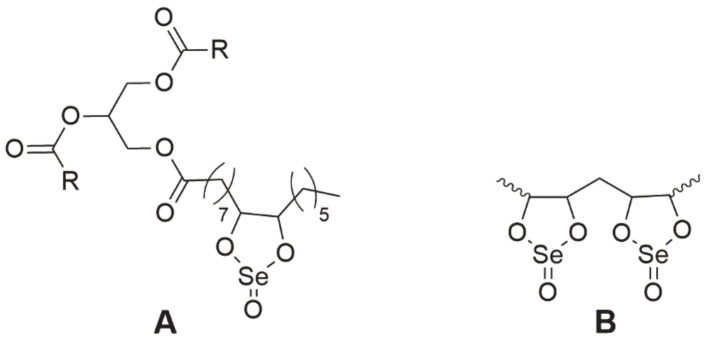
The chemical structure of Selol. **A**—2% Selol (single dioxaselenolane rings), **B**—5% Selol, and more (single and double dioxaselenolane rings). R, polyunsaturated fatty acid moiety.

**Figure 10 ijms-22-01009-f010:**
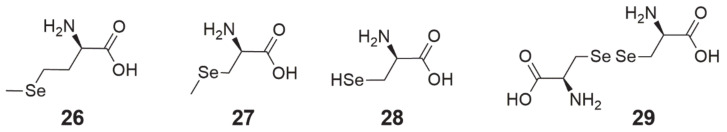
Natural selenoamino acids discussed in this paper (**26**—selenomethionine, **27**—methylselenocysteine, **28**—selenocysteine, **29**—selenocystine).

**Table 1 ijms-22-01009-t001:** Sodium selenite under clinical investigation. This table is compiled from the information available at https://www.clinicaltrials.gov/ on 7 January 2021.

NCT Number	Study Title	Clinical Trial Status	Study Design	References
NCT04296578	Study evaluating sodium selenite in combination with abiraterone in castration-resistant prostate cancer progressing on abiraterone	Phase 1 not yet recruiting	Non-randomized, open-label, single-group assignment	[40]
NCT01155791	Phase I sodium selenite in combination with docetaxel in castration-resistant prostate cancer	Phase 1 terminated	Open-label, single-group assignment	[41]
NCT02184533	Sodium selenite and radiation therapy in treating patients with metastatic cancer	Phase 1 completed	Open-label, single-group assignment	[42]
NCT00188604	The use of selenium to treat secondary lymphedema breast cancer	Phase 2 completed	Randomized, double-blind, placebo-controlled, crossover assignment	[43]
NCT01959438	Sodium selenite as a cytotoxic agent in advanced carcinoma (SECAR)	Unknown	Open-label, single-group assignment	[44]
NCT04201561	High dose inorganic selenium for preventing chemotherapy induced peripheral neuropathy (SELENIUM)	Phase 3 recruiting	Randomized, triple-blind (participant, care provider, investigator), placebo-controlled, parallel assignment	[45]

**Table 2 ijms-22-01009-t002:** Recent and current clinical trials of SeMet. This table is compiled from the information available at https://www.clinicaltrials.gov/ on 7 January 2021.

NCT Number	Study Title	Clinical Trial Status	Study Design	References
NCT00736164	Selenomethionine in treating patients undergoing surgery or internal radiation therapy for stage I or stage II prostate cancer	Phase 2 withdrawn	Randomized, double-blind, placebo-controlled	[106]
NCT00526890	Carboplatin, paclitaxel, selenomethionine, and radiation therapy in treating patients with stage III non-small cell lung cancer that cannot be removed by surgery	Phase 2 terminated	Open-label, single-group assignment	[107]
NCT00736645	Selenomethionine and finasteride before surgery or radiation therapy in treating patients with stage I or stage II prostate cancer	Phase 2 completed	Randomized, double-blind (participant, investigator), placebo-controlled, parallel assignment	[108]
NCT00030901	S9917, Selenium in preventing cancer in patients with neoplasia of the prostate	Phase 3 completed	Randomized, triple-blind (participant, care provider, investigator), placebo-controlled, parallel assignment	[109]
NCT00625183	Capecitabine, oxaliplatin, selenomethionine, and radiation therapy in treating patients undergoing surgery for newly diagnosed stage II or III rectal adenocarcinoma	Phase 2 terminated	Open-label, single-group assignment	[110]
NCT01211561	Colon cancer prevention using selenium	Unknown	Randomized, double-blind (participant, investigator), placebo-controlled, single group assignment	[111]
NCT00008385	Selenium in preventing tumor growth in patients with previously resected stage I non-small cell lung cancer	Phase 3 completed	Randomized, triple-blind (participant, care provider, investigator), placebo-controlled, parallel assignment	[112]
NCT00217516	Selenium in treating patients who are undergoing brachytherapy for stage I or stage II prostate cancer	Phase 1 completed	Non-randomized, open-label, single-group assignment	[113]
NCT00112892	Irinotecan and selenium in treating patients with advanced solid tumors	Phase 1 completed	Dose-escalation	[114]
NCT01497431	Se-methyl-seleno-l-cysteine or selenomethionine in preventing prostate cancer in healthy participants	Phase 1 completed	Randomized, double-blind (participant, investigator), placebo-controlled, parallel assignment	[115]
NCT01682031	Selenomethionine in reducing mucositis in patients with locally advanced head and neck cancer who are receiving cisplatin and radiation therapy	Phase 2 terminated	Randomized, double-blind, placebo-controlled, parallel assignment	[116]
NCT00006392	S0000 Selenium and vitamin E in preventing prostate cancer (SELECT)	Phase 3 completed	Randomized, quadruple -blind (participant, care provider, investigator, outcomes assessor), placebo-controlled, parallel assignment	[117]
NCT00706121	S0000D: Effect of vitamin E and/or selenium on colorectal polyps in men enrolled on SELECT trial SWOG-S0000 (ACP)	Phase 3 completed	Randomized, quadruple -blind (participant, care provider, investigator, outcomes assessor), placebo-controlled, parallel assignment	[118]
NCT02535533	SLM + axitinib for clear cell RCC	Phase 1/2 recruiting	Open-label, single-group assignment	[119]
NCT04683575	Clinical study on the effect of selenium yeast capsule on prognosis of differentiated thyroid carcinoma	Phase 4 not yet recruiting (new)	Randomized, double-blind (participant, investigator), placebo-controlled, parallel assignment	[120]

**Table 3 ijms-22-01009-t003:** Clinical trials using MSC. This table is compiled from the information available at https://www.clinicaltrials.gov/ on 7 January 2021.

NCT Number	Study Title	Clinical Trial Status	Study Design	References
NCT01611038	Chemoprevention of breast and prostate cancers in shift workers by dietary methylselenocysteine: effects on circadian rhythm and estrogen receptor-B cycling	Not applicable	Randomized, triple-blind (participant, investigator, outcomes assessor), placebo-controlled, parallel assignment	[123]
NCT01497431	Se-methyl-seleno-l-cysteine or selenomethionine in preventing prostate cancer in healthy participants	Phase 1 completed	Randomized, double-blind (participant, investigator), placebo-controlled, parallel assignment	[115]
NCT00829205	Se-methyl-seleno-l-cysteine, rituximab, ifosfamide, carboplatin, and etoposide in treating patients with diffuse large B-cell lymphoma that has relapsed or not responded to treatment	Phase 1/2 withdrawn	Non-randomized, open-label	[124]

## Data Availability

Not applicable.

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
