# Peer review of "Selenium Compounds as Novel Potential Anticancer Agents"

_ijms, 2021, doi:10.3390/ijms22031009_

Round 1
Reviewer 1 Report
In this manuscript, Radomska et al., reviewed the role of various of selenium compounds as potential novel anticancer agents. The manuscript is interesting; however, it lacks in vivo studies, especially data from human trials. You should include results (key findings) from Nutritional Prevention of Cancer Trial (NPC), and SELECT trial in your introduction or in the main text.
“All possible types of cell death induced by Se-containing compounds are presented/summarised in Figure 1”, you need to include Ferroptosis in this Figure, and discuss the role of Se and Glutathione peroxidase especially GPX4 and Ferroptosis.
You may include some side-effects of Se such as high selenium intake (via supplementations) may increase the incidence of Type II diabetes, there are many papers in the literature.
Author Response
Thank you for your review of our manuscript entitled “Selenium compounds as novel potential anticancer agents” for publication in International Journal of Molecular Sciences. Overall, we find Reviewer suggestions to be helpful, and constructive, and the corresponding revisions have strengthened the paper in multiple ways. According to your suggestions, we have expanded our manuscript with the content you suggested (the changes made in the manuscript are marked in red).
Point 1: The manuscript is interesting; however, it lacks in vivo studies, especially data from human trials.
Response 1: As suggested by the Reviewer, we added in vivo studies (human trails) to the manuscript. These data are presented in the form of tables.
Point 2: You should include results (key findings) from Nutritional Prevention of Cancer Trial (NPC), and SELECT trial in your introduction or in the main text.
Response 2: Due to the breadth and complexity of the issues surrounding selenium, we wanted to devote only to the description of its anticancer properties. Furthermore, due to the numerous papers on the NPC and SELECT trials, our attention was directed only to Se compounds anticancer properties. At the same time, taking into account the reviewer's remarks, we are in the process of preparing an article that will include studies concerning the above mentioned issue.
Point 3: “All possible types of cell death induced by Se-containing compounds are presented/summarised in Figure 1”, you need to include Ferroptosis in this Figure, and discuss the role of Se and Glutathione peroxidase especially GPX4 and Ferroptosis.
Response 3: As suggested by Reviewer, we have added ferroptosis (and other types of cell death caused by Se compounds) to Figure 1. In addition, we have added a description of ferroptosis and the role of Se and Gpx4 to the main text (page 2, line 65 - 88).
Point 4: You may include some side-effects of Se such as high selenium intake (via supplementations) may increase the incidence of Type II diabetes, there are many papers in the literature.
Response 4: Similar to the NPC and SELECT trails (Point 2), side-effects of Se will be described and discussed in our next manuscript (Response 2) - we are in the process of its preparation.
We hope that our answer will satisfy Reviewer and Editor. We thank you for consideration and await your decision.

Reviewer 2 Report
Manuscript entitled “Selenium compounds as novel potential anticancer agents” presents the study of anticancer activity of selenium-containing compounds. This manuscript presents a less extensive but different approach to this issue than the reviews on the subject cited by the authors (ref. 5 and 8). The work is written in a concise and interesting way. In my opinion, the paper should be published after minor revision.
I suggest the following corrections:
- The authors should correct figure 3 - no bonds between Se atoms are visible. In order to improve the aesthetics of the manuscript, the authors should standardize the figures - the same size of the rings, the length of the bonds and the numbers of substances in the same font.
- Figure 4 is not entirely clear what A and B mean? Why did the authors introduce this?
- The authors sometimes report IC50 values with standard deviation, other times they do not. It would be good to standardize it.
- There are also some editing errors in the manuscript:
- page 6, line 225 – CA-4 and 16 should be bolded,
- page 6, line 240 – Fig. 5, structure 17 should be bolded,
- page 7, line 271 – Fig. 5, structure 18 should be bolded,
- page 15, line 639/640 – Fig. 10, structure 28 should be bolded.
It is incomprehensible to cite reference number 29 - this manuscript does not concern selenium compounds.
Author Response
Thank you for your review of our manuscript entitled “Selenium compounds as novel potential anticancer agents” for publication in International Journal of Molecular Sciences. Overall, we find Reviewer suggestions to be helpful, and constructive, and the corresponding revisions have strengthened the paper in multiple ways. According to your suggestions, we have expanded our manuscript with the content you suggested (the changes made in the manuscript are marked in red).
Point 1: The authors should correct figure 3 - no bonds between Se atoms are visible. In order to improve the aesthetics of the manuscript, the authors should standardize the figures - the same size of the rings, the length of the bonds and the numbers of substances in the same font.
Response 1: As suggested by the Reviewer, Figure 3 has been corrected. We have sent the corrected Figures to the Editor (the Author's Reply to the Review Report template does not have an option to resubmit corrected Figures).
“… the authors should standardize the figures - the same size of the rings, the length of the bonds and the numbers of substances in the same font”- the size of the chemical formulas are dependent on the size of the inserted image in the text file.
Point 2: Figure 4 is not entirely clear what A and B mean? Why did the authors introduce this?
Response 2: In order to enable the Reader to follow of the mechanism occurring during molecular docking of the compound 16 (description in text), we introduced the A and B ring designations in Figure 4. Having regard to the Reviewer's commentsin, in order for the reader to follow the above mechanism in a simpler way signages A and B were added to a main text (page 8, line 313, 315).
Point 3: The authors sometimes report IC50 values with standard deviation, other times they do not. It would be good to standardize it.
Response 3: As suggested by the Reviewer, IC50 values have been standardized.
Point 4: There are also some editing errors in the manuscript:
page 6, line 225 – CA-4 and 16 should be bolded,
page 6, line 240 – Fig. 5, structure 17 should be bolded,
page 7, line 271 – Fig. 5, structure 18 should be bolded,
page 15, line 639/640 – Fig. 10, structure 28 should be bolded.
Response 4: As suggested by the Reviewer, all of the above phrases are in bold.
Point 5: It is incomprehensible to cite reference number 29 - this manuscript does not concern selenium compounds.
Response 5: Cite reference number 29 has been deleted.
We hope that our answer will satisfy Reviewer and Editor. We thank you for consideration and await your decision.

Reviewer 3 Report
In this manuscript, the authors comprehensively and concisely review the novel anticancer effects of selenium compounds. Indeed, the authors cite several recent publications in the manuscript. I think that the manuscript is worthy for publication in International Journal of Molecular Sciences, and is expected to be highly cited by future articles. However, there are some points which should be reconsidered before publication.
line 63: “All possible types of cell death induced by Se-containing compounds are presented in Figure 1.” Nowadays, many types of cell death are known in addition to apoptosis, autophagy and necrosis, e.g., necroptosis, pyroptosis, ferroptosis and so on. In Figure 1, few classical types of cell death were mentioned. The description seems to be misleading.
line 154: Diselenides were actually effective against cultured cancer cell lines. Besides, they could have severe toxicity in vivo, and are unapplicable to cancer chemotherapy due to their toxicity. The authors should discuss the toxicity of the compounds and evaluate the risk and the benefit.
line 397: The authors mentioned that selenoesters were hydrolyzed by an enzyme (probably esterase). Have the esterase which can hydrolyze selenoesters been identified?
Author Response
Thank you for your review of our manuscript entitled “Selenium compounds as novel potential anticancer agents” for publication in International Journal of Molecular Sciences. Overall, we find Reviewer suggestions to be helpful, and constructive, and the corresponding revisions have strengthened the paper in multiple ways. According to your suggestions, we have expanded our manuscript with the content you suggested (the changes made in the manuscript are marked in red).
Point 1: line 63: “All possible types of cell death induced by Se-containing compounds are presented in Figure 1.” Nowadays, many types of cell death are known in addition to apoptosis, autophagy and necrosis, e.g., necroptosis, pyroptosis, ferroptosis and so on. In Figure 1, few classical types of cell death were mentioned. The description seems to be misleading.
Response 1: To avoid misleading the reader, the sentence has been corrected to: “Possible types of cell death induced by Se-containing compounds are presented in Figure 1” (page 2, line 87). In addition, other types of cell death suggested by the Reviewer that were indeed induced by Se compounds and found in recent literature were added.
Point 2: line 154: Diselenides were actually effective against cultured cancer cell lines. Besides, they could have severe toxicity in vivo, and are unapplicable to cancer chemotherapy due to their toxicity. The authors should discuss the toxicity of the compounds and evaluate the risk and the benefit.
Response 2: As suggested by the Reviewer, we discussed the toxicity of this class of compounds, which have been described on in vivo models in the available literature (page 7 and 8, lines 244 - 292). The change suggested by the Reviewer, in our opinion, has greatly enriched our work and broadened its horizons.
Point 3: line 397: The authors mentioned that selenoesters were hydrolyzed by an enzyme (probably esterase). Have the esterase which can hydrolyze selenoesters been identified?
Response 3: According to the current state of knowledge, no specific enzyme has been identified that is only responsible for the hydrolysis of these compounds. In accordance to data from studies selenoesters can be reduced by enzyme such as TrxR1 or can be hydrolyzed by nucleophiles such as water. We also found articles that suggested that carbonic anhydrases or acetylcholinesterase (AchE) may be responsible for hydrolyzing these compounds. However, it turns out that AchE is a non-specific enzyme (it degrades not only Ach) and can hydrolyze selenoesters to simpler compounds. The above information has been included in the main text in the subsection "Selenoesters".
We hope that our answer will satisfy Reviewer and Editor. We thank you for consideration and await your decision.

Round 2
Reviewer 1 Report
It was disappointing that two Trials (NPC and SELECT) on se are not discussed.
The authors have addressed most of the comments, I can recommend to accept this revised version for publication.